# UPLC-ESI-MS/MS-Based Widely Targeted Metabolomics Analysis of Wood Metabolites in Teak (*Tectona grandis*)

**DOI:** 10.3390/molecules25092189

**Published:** 2020-05-07

**Authors:** Guang Yang, Kunnan Liang, Zaizhi Zhou, Xiyang Wang, Guihua Huang

**Affiliations:** Research Institute of Tropical Forestry, Chinese Academy of Forestry, Guangzhou 510520, China; ritfkyc@caf.ac.cn (G.Y.); lkn@ritf.ac.cn (K.L.); zzzhou@ritf.ac.cn (Z.Z.); teakpro@126.com (X.W.)

**Keywords:** *Tectona grandis*, UPLC–ESI–MS/MS-based metabolomics, sapwood, heartwood

## Abstract

The properties of teak wood, such as natural durability and beautiful color, are closely associated with wood extractives. In order to further understand the performance differences between teak heartwood and sapwood, we analyzed the chemical components of extractives from 12 wood samples using an ultrahigh-performance liquid chromatography-electrospray ionization-tandem mass spectrometry (UPLC-ESI-MS/MS)-based metabolomics approach. In total, 691 metabolites were identified, and these were classified into 17 different categories. Clustering analysis and principal component analysis of metabolites showed that heartwood samples could be clearly separated from sapwood samples. Differential metabolite analysis revealed that the levels of primary metabolites, including carbohydrates, amino acids, lipids, and nucleotides, were significantly lower in the heartwood than in the sapwood. Conversely, many secondary metabolites, including flavonoids, phenylpropanoids, and quinones, had higher levels in the heartwood than in the sapwood. In addition, we detected 16 specifically expressed secondary metabolites in the heartwood, the presence of which may correlate with the durability and color of teak heartwood. Our study improves the understanding of differential metabolites between sapwood and heartwood of teak, and provides a reference for the study of heartwood formation.

## 1. Introduction

Teak (*Tectona grandis* L.F.) is a tropical hardwood tree species belonging to the Lamiaceae family. It is native to India, Myanmar, Laos, and Thailand, and is widely naturalized and cultivated in tropical and subtropical countries [1]. Teak is a high-value timber tree, mainly owing the beautiful texture and natural durability of the timber. These remarkable properties have resulted in its utilization in a wide range of applications, such as house construction, outdoor furniture, and flooring [2].

Natural durability is an important property to consider when evaluating the quality and value of timber, and this is closely related to its chemical composition, especially the presence of wood extractives [3,4,5]. Heartwood durability exhibits a high degree of variability in teak. Juvenile and plantation fast-growth teak wood have lower natural durability than mature and naturally grown teak owing to their low extractive content [6,7,8]. In addition, teak sapwood is less durable than teak heartwood, mainly owing to the presence of non-structural carbohydrates and the low extractive content of sapwood [9]. Previous studies have indicated that the natural durability of teak wood can be ascribed to the presence of toxic extractives, such as anthraquinones and naphthoquinones. Some studies have shown that the main substance that influences the natural durability of teak wood is tectoquinone [8,10,11]. Another study indicated that P1,2-(hydroxymethyl) anthraquinone and tectoquinone could be responsible for the natural durability of teak [9]. However, another study suggested that 1,4-naphthoquinone is the most important compound responsible for the natural durability of teak [12]. In conclusion, the substances responsible for the natural durability of teak remain unclear, and warrant more in-depth investigation.

The extractives not only determine the durability of the wood but can also affect the color of the wood. Color is a key quality criterion for wood since it determines the commercial value and suitability for a given product [13]. Teak heartwood is valued for its attractive appearance and has a golden yellow or brown color [14]. The CIELab(parameters *L**, *a**, and *b**) system has been used to determine wood color, and it was found that the heartwood color of teak varies widely between different trees. The heartwood color of wood from fast-growth plantations is lighter than that of wood from native trees [15], because heartwood color is relatively darker in wetter than drier areas [14]. In addition to the presence of color variation in teak heartwood between different trees, there are more obvious color changes between the sapwood and heartwood within an individual tree [6,16,17]. In recent years, researchers have attempted to determine the causes of wood coloration. Gierlinger [18] found that the change in color from larch (*Larix* spp.) sapwood to heartwood was due to the synthesis and accumulation of extractives during heartwood formation. Similar results were found in teak, and the extractive content in organic solvents showed the rise from sapwood to heartwood that is typical in plantation teakwood from Panama [19]. A more recent study also found that the amount of acetone extractives in the heartwood of teak was higher than that in the sapwood, especially for phenols, quinones, and ketones [4]. Researchers have speculated that the obvious differences in substances between the heartwood and sapwood may result in the chromatic aberrations observed in teak [4]. Nonetheless, there is limited information about the substances that affect the color of teak wood, and further study is therefore required.

Although extensive research has been undertaken into the extractive content, including the types of extractives, few studies have been conducted on the extractive chemical composition of teak [3,9,19,20]. Quinones are a major class of extract, and they are present in teak in the form of naphthoquinones and anthraquinones [21]. Tectoquinone was the first compound to be isolated from the extract [22], and since then researchers have successively isolated several quinones from different tissues of teak [23,24,25,26,27]. Currently, 92 metabolites, including quinones, terpenoids, apocarotenoids, phenolics, and flavonoids, have been isolated from the root, leaf, and heartwood of teak [21]. However, information on the amount and function of these chemicals is still very limited and needs to be further explored.

Widely targeted metabolomic analysis is a novel approach that combines the advantages of non-targeted metabolomics and targeted metabolomics. Widely targeted metabolomics obtains the potential targets, measured by screening the samples using multiple reaction monitoring (MRM) conditions optimized from the available authentic compounds. Compared to total scan ESI-based non-targeted metabolomics, widely targeted metabolomics based on MRM is a very sensitive and accurate method for the measurement of targeted metabolites [28]. In recent years, ultrahigh-performance liquid chromatography-electrospray ionization-tandem mass spectrometry (UPLC-ESI-MS/MS)-based widely targeted metabolomic methods have been applied to plant metabolomic analyses [28,29,30]. In the present study, a widely targeted metabolomic method was employed to analyze the types and relative content of metabolites of the heartwood and sapwood of teak, and to compare the two. We believe that our findings will facilitate a deeper understanding of differential metabolites between sapwood and heartwood and provide a reference for future study on the heartwood formation mechanism.

## 2. Materials and Methods

### 2.1. Plant Materials

Wood samples were collected in the form of cores (5 mm diameter) using a hand-driven core borer from the stems of nine 18-year-old trees of teak clones grown in Jieyang, Guangdong Province, southern China, in October 2018. Cores were drilled in each tree trunk at approximately breast height level (130 cm). The core samples were immediately submerged in liquid nitrogen and stored at −80 °C in a laboratory freezer. In the laboratory, the core was divided into four parts: the heartwood zone (HWZ), which is the central portion of the stems near the pith; the interior sapwood zone (ISZ), which is next to the HWZ; the middle sapwood zone (MSZ); and the exterior sapwood zone (ESZ), which is close to the cambium. The sample from each zone was prepared using three biological replicates (Figure 1).

### 2.2. Sample Preparation

The cores were crushed using a mixer mill (MM 400, Retsch, Haan, Germany) with liquid nitrogen for 30 s at 30 Hz, and this was repeated four times. Then, 100 mg freeze-dried powder was extracted overnight at 4 °C using 1.0 mL 70% aqueous methanol. Following centrifugation at 10,000 × *g* for 10 min, the extracts were absorbed and filtrated. The obtained filtrates were subjected to liquid chromatography-tandem mass spectrometry (LC-MS/MS) analysis. In addition, a quality control (QC) sample was prepared by mixing all the core samples. During the instrumental analysis, a QC sample was inserted alongside each of the 10 test samples to examine the repeatability of the analytical process.

### 2.3. HPLC Conditions

To investigate metabolite changes in the HWZ and the sapwood zone (SWZ), all wood samples (HWZ, ISZ, MSZ, and ESZ) were investigated with a widely targeted metabolomics method using an LC-ESI-MS/MS system (HPLC, Shim-pack UFLC SHIMADZU CBM30A system, MS, Applied Biosystems 6500 Q TRAP). The analytical conditions were based on Wang et al. [31]. HPLC: column, Waters ACQUITY UPLC HSS T3 C18 (1.8 µm, 2.1 mm × 100 mm, Milford, MA, USA,); solvent system, water (0.04% acetic acid): acetonitrile (0.04% acetic acid); gradient program, 95:5 *v/v* at 0 min, 5:95 *v/v* at 11.0 min, 5:95 *v/v* at 12.0 min, 95:5 *v/v* at 12.1 min, and 95:5 *v/v* at 15.0 min; flow rate, 0.40 mL/min; temperature, 40 °C; injection volume: 2 μL. The effluent was connected to an ESI-triple quadrupole-linear ion trap (Q TRAP)-MS.

### 2.4. ESI-Q TRAP-MS/MS

The LC-ESI-MS/MS analysis was performed by Metware Biotechnology Co., Ltd. (Wuhan, China). Mass spectrometry followed the method of Wang et al. [31]. Linear ion trap (LIT) and triple quadrupole (QQQ) scans were acquired using a triple quadrupole-linear ion trap mass spectrometer (Q TRAP), API 6500 Q TRAP LC-MS/MS System, equipped with an ESI Turbo Ion-Spray interface, operating in both positive and negative ion mode, and controlled by Analyst 1.6.3 software (AB Sciex, Framingham, MA, USA). The ESI source operation parameters were as follows: ion source, turbo spray; source temperature 500 °C; ion spray voltage 5500 V; ion source gas I (GSI), gas II (GSII), and curtain gas (CUR) were set at 55, 60, and 25.0 psi, respectively; the collision gas (CAD) was high. Instrument tuning and mass calibration were performed with 10 and 100 μmol/L polypropylene glycol solutions in QQQ and LIT modes, respectively. QQQ scans were acquired as multiple reaction monitoring (MRM) experiments with collision gas (nitrogen) set to 5 psi. The declustering potential (DP) and collision energy (CE) for individual MRM transitions were carried out with further DP and CE optimization. A specific set of MRM transitions was monitored for each period according to the metabolites eluted within that period.

### 2.5. Qualitative and Quantitative Analysis of Metabolites

Qualitative and quantitative analyses of metabolites were undertaken using the methods of Wang et al. [31]. The qualitative analysis of primary and secondary mass spectrometry data was performed based on the self-built database MWDB (Metware Biotechnology Co., Ltd. Wuhan, China) and the publicly available metabolite databases. The interference from isotope signals; repeated signals of K^+^, Na^+^, and NH4 ^+^ ions; and fragment ions derived from other larger molecules were eliminated during identification. Metabolite structure analysis was obtained by referencing existing mass spectrometry databases such as MassBank (http://www.massbank.jp), KNAPSAcK (http://kanaya.naist.jp/KNApSAcK), and METLIN (http://metlin.scripps.edu/index.php).

The quantitative analysis of metabolites was performed using MRM mode of QQQ mass spectrometry. In the MRM mode, the quadrupole filters the precursor ions of the target substance and excludes the ions corresponding to other molecular weights to eliminate interference. After obtaining metabolite mass spectrometry data, the mass spectrum peaks were subjected to integration and correction using MultiQuant version 3.0.2 (AB SCIEX, Concord, Ontario, Canada). Finally, the corresponding metabolite contents were represented as chromatographic peak area integrals.

### 2.6. Statistical Analysis

Principal component analysis (PCA) [29] and orthogonal projections to latent structures-discriminant analysis (OPLS-DA) [32] were performed on the metabolic data of all samples to assess the metabolite diversity between and within group samples. The data were normalized prior to analysis. Two screening criteria for significant differential metabolites were established: a fold change of ≥2 or of ≤0.5 and *p*-value <0.05 (Student’s *t*-test); and the variable importance in the projection (VIP) in the OPLS-DA model of ≥1. Visualization of the metabolites was performed using cluster analysis.

## 3. Results

### 3.1. Widely Targeted Metabolic Profiling of Teak Wood

The total ion current (TIC) plot of one QC sample shows the summed intensity of all ions in the mass spectrum at different time points (Appendix A). The stacking diagram of the TIC plot from QC mass spectrometry (Appendix A) and the TIC curve of the metabolites showed high overlap, thus demonstrating data repeatability and reliability. The multipeak detection plot of metabolites in multiple reaction monitoring (MRM) mode shows the ion current plot of multiple substances (Appendix A). In the MRM plot, each mass spectrum peak in a different color represents a detected metabolite. A total of 691 metabolites were detected, including: 125 flavonoids, 100 organic acids and derivatives, 78 amino acids and derivatives, 66 phenylpropanoids, 61 lipids, 57 nucleotides and derivates, 40 alkaloids, 27 terpenes, 22 carbohydrates, 18 vitamins and derivatives, 18 alcohols, 14 phenolamides, 11 polyphenols, six quinones, five sterides, five indole derivatives, and 38 others (Appendix A, Figure 2a). In the clustering heat map, the accumulation of metabolites displayed clear phenotypic variation in terms of the pattern of metabolite abundance in different tissues, and the HWZ was clearly distinct from the SWZs (ISZ, MSZ, and ESZ) (Figure 2b).

### 3.2. PCA for All Samples

A multivariate statistical analysis by PCA of the metabolite profile of 12 wood samples was performed. Two principal components, PC1 and PC2, were extracted and explained 47.66% and 22.44% of the variability, respectively. In the PCA score plot (Figure 3a), the three biological replicates for each sample clustered, indicating that there was little variation among core samples within groups, and the experiment was reproducible and reliable. The core samples of the different zones were clearly separated, suggesting that there were significant differences in metabolic phenotypes between them.

### 3.3. OPLS-DA for all Samples

The OPLS-DA model compared the entire metabolite content of the core samples in pairs to evaluate the differences between HWZ and ISZ (R2X = 0.785, R2Y = 1, Q2Y = 0.988; Figure 3b), HWZ and MSZ (R2X = 0.832, R2Y = 1, Q2Y = 0.993; Figure 3c), and HWZ and ESZ (R2X = 0.862, R2Y = 1, Q2Y = 0.997; Figure 3d). The Q2 values of all comparison groups exceeded 0.9, thus demonstrating that these models were stable and reliable and could be used to further screen for differential metabolites.

### 3.4. Differential Metabolite Screening

Differential metabolites were screened for each comparison group by combining the fold change and VIP values of the OPLS-DA model. The screening results of an analysis run with a VIP of ≥1.0 and fold changes of ≥2 or ≤ 0.5 as thresholds for significant differences are shown in Figure 4 and Appendix A. There were 332 significantly different metabolites between HWZ and ISZ (ISZ had 113 down-regulated and 219 up-regulated), 370 between HWZ and MSZ (MSZ had 129 down-regulated and 241 up-regulated), and 413 between HWZ and ESZ (ESZ had 145 down-regulated and 268 up-regulated) (Figure 4a). The results showed that with the increase in distance from HWZ, the amounts of differential metabolites increased. In addition, there were more up-regulated metabolites than down-regulated metabolites in the SWZs (ISZ, MSZ, and ESZ) than in the HWZ. Furthermore, 247 common differential metabolites were found in the comparison groups HWZ vs. ISZ, HWZ vs. MSZ, and HWZ vs. ESZ (Figure 4b, Appendix A). These common differential metabolites were classified into 16 different categories, including: 47 amino acids and derivatives, 33 organic acids and derivatives, 30 nucleotide and derivates, 28 flavonoids, 24 lipids, 21 phenylpropanoids, 12 carbohydrates, 12 alkaloids, 11 terpenes, seven vitamins and derivatives, four quinones, four polyphenols, two indole derivatives, one phenolamide, one alcohol, and 10 others (Figure 4c).

### 3.5. Differential Primary Metabolites Analysis

A total of 154 common differential primary metabolites were found in the comparison groups HWZ vs. ISZ, HWZ vs. MSZ, and HWZ vs. ESZ. Analysis revealed that the levels of most of these, including carbohydrates, amino acids, lipids, and nucleotides, in the HWZ were significantly lower than those in the SWZ. A total of 12 common differential carbohydrates were detected in the three comparison groups (HWZ vs. ISZ, HWZ vs. MSZ, and HWZ vs. ESZ), of which the levels of d-Fructose 6-phosphate, Glucose-1-phosphate, and d-Fructose 6-phosphate-disodium salt were the lowest in the ESZ, whereas the levels of the other carbohydrates were lower in the HWZ than in the SWZ (Figure 5a). We detected 24 differential lipids, and the levels of most of these decreased gradually from ESZ to HWZ; however, 4-hydroxysphinganine, α-linolenic acid, and 12,13-EODE had higher levels in the HWZ (Figure 5b). Furthermore, 47 differential amino acid and derivatives were detected; with the exception of d-erythro-sphinganine, the level of other amino acids was significantly lower in the HWZ compared to the SWZ (Figure 5c). In addition, 30 differential nucleotide and derivates were detected, of which the levels of UDP-α-d-glucose and uridine 5ʹ-diphosphoglucose disodium salt in the ESZ were the lowest; however, the others had lower levels in the HWZ compared to the SWZ (Figure 5d).

### 3.6. Differential Secondary Metabolites Analysis

We found 93 common differential secondary metabolites among the comparison groups HWZ vs. ISZ, HWZ vs. MSZ, and HWZ vs. ESZ. Of these, flavonoids, phenylpropanoids, terpenes, and quinones accounted for a high proportion of the metabolites. A total of 28 flavonoid metabolites, including nine flavones, six flavonols, four isoflavones, three flavanones, and six flavonoids were detected, and the levels of 20 of these flavonoids were significantly higher in the HWZ than in the SWZ (Figure 6a). We also detected 21 phenylpropanoids, and the levels of 13 of these were significantly higher in the HWZ than in the SWZ (Figure 6b). A total of four different quinones were detected in this study; of these, the levels of rhein, aloe emodin, and emodin were significantly higher in the HWZ than in the SWZ, and rhein was only detected in the HWZ. Interestingly, another quinone, chrysophanic acid, was only detected in the SWZ (Figure 6c). In addition, five of 11 terpenes and three of 12 alkaloids were detected, and their levels were higher in the HWZ (Figure 6d,e).

In order to analyze the useful properties of the HWZ, we examined the HWZ-specific metabolites. Based on the expression analysis of secondary metabolites among the comparison groups HWZ vs. ISZ, HWZ vs. MSZ, and HWZ vs. ESZ (Appendix A), a total of 16 metabolites could only be detected in heartwood, and these were referred to as HWZ-specific metabolites. The integral diagram of 16 metabolites is showed in Appendix A. These included three phenylpropanoids, eight flavonoids, one terpene, one quinone, one alkaloid, and two others (Table 1).

## 4. Discussion

### 4.1. Wide Target Metabolome

Extractives account for small parts of wood, but greatly influence the properties of wood. Several studies have shown that extractives are one of the most important factors affecting the durability and color of teak wood [6,7,8]. Previous research on the extractives of teak has mostly focused on the content and type of extract; consequently, information on the identification and isolation of the composition of extractives is scarce. Recently, 49 and 29 metabolites from the HWZ and SWZ of teak, respectively, were detected in an acetone extractive using gas chromatography–mass spectrometry (GC-MS) [4]. In the present study, a total of 691 metabolites were detected in teak wood core samples using a widely targeted metabolomics method based on LC-ESI-MS/MS. Our study obtained substantial information on metabolites of teak wood and provides a basis for the separation and traditional use of valuable components.

### 4.2. Differential Metabolites

Teak is a precious timber species, and it has obvious differences in its physical properties; for instance, durability and color differ between the SWZ and HWZ [6,16]. A clustering heat map of the metabolites showed that the HWZ can be clearly distinguished from three SWZ, suggesting that there are significant differences in metabolic phenotypes between them (Figure 2b). An analysis of differential metabolites also found that the expression pattern of metabolites from the three SWZ (ISZ, MSZ, and ESZ,) was similar, and that they were obviously different from the HWZ. Therefore, analysis of differential metabolites is useful for understanding the differences in properties between the HWZ and SWZ.

Differential metabolite analysis found that the levels of primary metabolites, including carbohydrates (soluble sugars), amino acids, lipids, and nucleotides, in the HWZ were significantly lower than those in the SWZ (Figure 5). This result is in agreement with those of previous histochemical studies [33,34], and may be because sapwood contains living parenchyma cells, whereas heartwood is generally considered to be dead tissue. Non-structural carbohydrates (soluble sugars and starches) and lipids serve as reserve materials in the sapwood [35], which provide energy not only for the metabolism of parenchymal cell but also for water conduction [36]. The heartwood, a dead tissue, is gradually transformed from sapwood. Current metabolite data indicate that the amount of differential metabolites increases as the distance from the HWZ increases. During the heartwood formation process, the high sapwood reserve material content is utilized by converting it into heartwood, resulting in the scarcity of these reserve materials in the HWZ [9,34,37,38]. Consequently, carbohydrates and lipids are likely to be the major source of carbon skeletons for the synthesis of heartwood substances [36,39,40], and amino acids serve as precursors for the biosynthesis of secondary metabolites. The formation of the heartwood substances, concomitant with the consumption of reserve materials, nucleus disintegration, and chromatin loss, finally causes the death of parenchyma cells.

A characteristic feature of the HWZ is the accumulation of secondary metabolites. In the present study, the flavonoids, phenylpropanoids, and quinones were found to be significantly enriched in the HWZ (Figure 6). These heartwood substances (secondary metabolites) may give the wood its unique properties, such as natural durability and color [3,4,5]. In teak, natural durability is ascribed to the presence of toxic extractives, which are mostly quinones [9]. To date, several quinones responsible for the durability of teak wood have been detected, for instance tectoquinone [8,10,11], P1,2-(hydroxymethyl) anthraquinone [9], and 1,4-naphthoquinone [12]. In the present study, three previously unmentioned anthraquinone substances, rhein, emodin, and aloe emodin, were detected with high expression levels in teak HWZ (Figure 6c). These substances are major compounds found in rhubarb, and they have been investigated for anti-inflammatory, antibacterial, and antitumor activity [41,42,43,44]. Whether they are related to heartwood durability requires further experimental verification.

In teak, the heartwood is characterized by a darker color than the sapwood. This is attributed to the process of heartwood formation, with the extractives produced accumulating in the HWZ [36]. Studies have reported that the color-related compounds in wood, such as pigments, tannins, and resins, had phenolic hydroxyl groups, carbonyl groups, and double bond structures [45]. However, the substances responsible for the coloration of heartwood are not well understood. In a recent study, researchers speculated that 4-tert-butyl-2-phenyl-phenol, 2-methylanthraquinone, and 2,3-dimethyl-1,4,4a,9a-tetrahydro-9,10-anthracenedione might be the main compounds responsible for the HWZ being darker in color than the SWZ. Our study detected 16 specifically expressed metabolites in the HWZ, including three phenylpropanoids, eight flavonoids, and one quinone (Table 1), which might be related to the chromatic properties of teak wood. Our metabolite data suggest new substances that may affect heartwood color and provide a reference for future study on the chromatic properties of teak wood.

## 5. Conclusions

The present study represents the first attempt to analyze the chemical components of extractives from teak wood core samples using a widely targeted metabolomics method. Using this method, we identified 691 metabolites in teak wood. Differential metabolites analysis showed that high levels of flavonoid and phenylpropanoid metabolites accumulated in the HWZ, whereas carbohydrates, amino acids, lipids, and nucleotides were significantly lower in the HWZ than in the SWZ. A total of 16 specifically expressed secondary metabolites that may be associated with durability and color in the HWZ were detected. Our study provides a reference for the study of heartwood formation in the future.

## Figures and Tables

**Figure 1 molecules-25-02189-f001:**
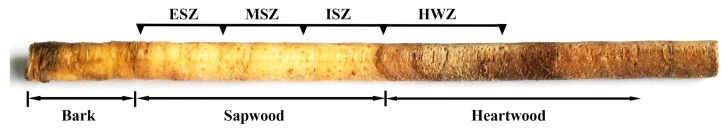
One of the teak wood core samples. Heartwood zone (HWZ), interior sapwood zone (ISZ), middle sapwood zone (MSZ), exterior sapwood zone (ESZ).

**Figure 2 molecules-25-02189-f002:**
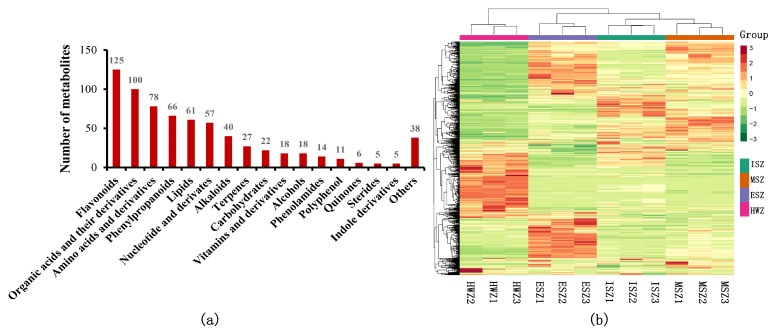
(**a**) Classification of detected metabolites. (**b**) Clustering heat map of the metabolites. The metabolite content data were normalized. Each sample is represented by a column, and each metabolite is represented by a row. A red bar indicates high abundance, whereas a green bar indicates a low relative abundance (the color key scale is beside the heat map).

**Figure 3 molecules-25-02189-f003:**
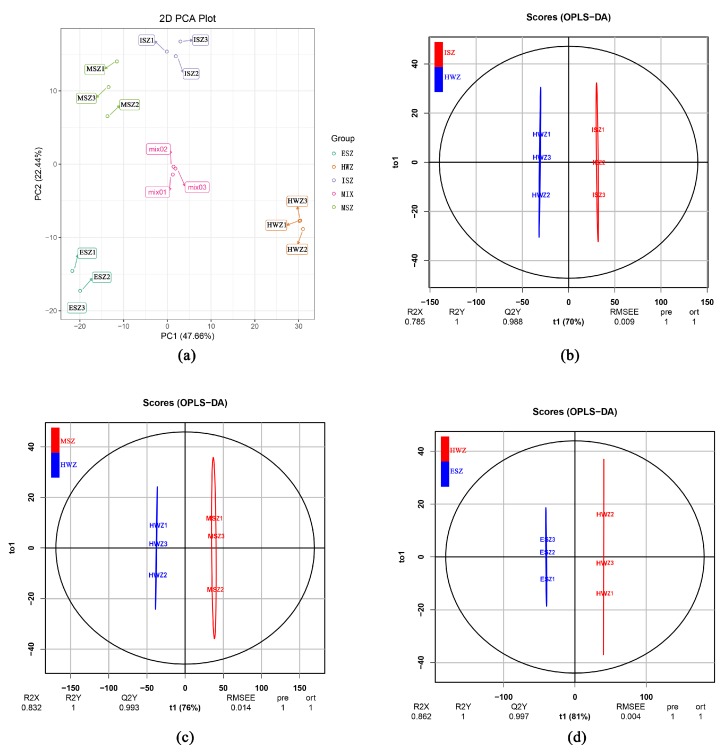
Differential metabolite analysis using principal component analysis (PCA) and orthogonal projections to latent structures-discriminant analysis (OPLS-DA). (**a**) PCA score plots. (**b**–**d**) OPLS-DA model plots for the comparison groups HWZ vs ISZ (**b**), HWZ vs MSZ (**c**), and HWZ vs ESZ (**d**).

**Figure 4 molecules-25-02189-f004:**
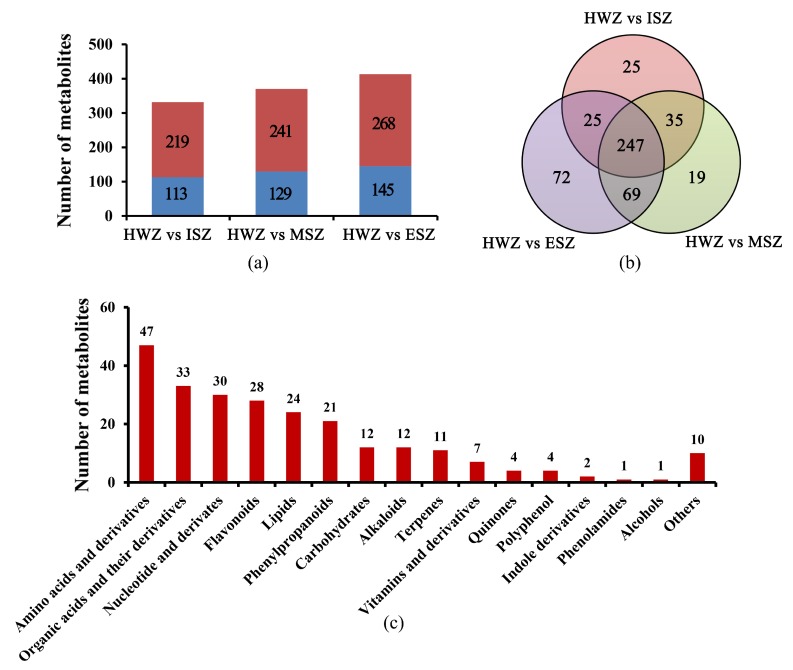
The differential metabolites analysis. (**a**) The number of differential metabolites between ISZ, MSZ, ESZ, and HWZ, respectively. Red pillars represent up-regulated differentially expressed metabolites; Blue pillars represent down-regulated differentially expressed metabolites. (**b**) The Venn diagram of differential metabolites of the comparison groups HWZ vs. ISZ, HWZ vs. MSZ, and HWZ vs. ESZ, respectively. (**c**) Number of different types of differential metabolites.

**Figure 5 molecules-25-02189-f005:**
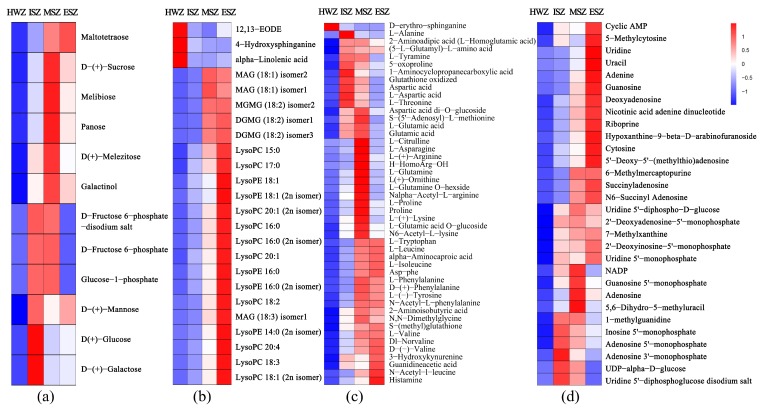
The differential primary metabolites analysis between ISZ, MSZ, ESZ, and HWZ. (**a**) carbohydrates, (**b**) amino acids, (**c**) lipids, and (**d**) nucleotides. Ratios of fold changes are given by shades of red or blue colors. Data represent mean values of three biological replicates for each sample.

**Figure 6 molecules-25-02189-f006:**
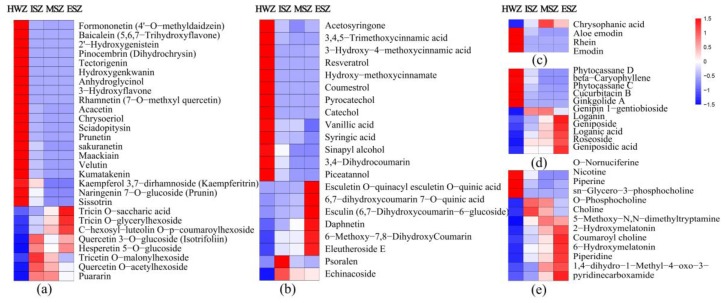
The differential secondary metabolites analysis between ISZ, MSZ, ESZ, and HWZ. (**a**) flavonoids, (**b**) phenylpropanoids, (**c**) quinones, (**d**) terpenes, and (**e**) alkaloids. Ratios of fold changes are given by shades of red or blue colors. Data represent mean values of three biological replicates for each sample.

**Table 1 molecules-25-02189-t001:** A list of HWZ-specific metabolites.

Id	Q1 (Da)	Q3 (Da)	Rt (min)	Molecular Weight (Da)	Compounds	Class
pmb0423	195.1	145.3	3.22	194.1	Hydroxy-methoxycinnamate	Phenylpropanoids
pme0307	227.1	185	4.78	228.08	Resveratrol	Phenylpropanoids
pmf0117	266.9	211.1	5.75	268.04	Coumestrol	Phenylpropanoids
pme1510	269.1	251	5.94	270.05	Baicalein (5,6,7-Trihydroxyflavone)	Flavone
pme3137	239	165	7.73	238.06	3-Hydroxyflavone	Flavonol
pme3369	315	165	6.43	316.06	Rhamnetin (7-O-methxyl quercetin)	Flavonol
pme2979	255.1	151	7.04	256.07	Pinocembrin (Dihydrochrysin)	Flavanone
pme3279	287	153	4.89	286.05	2’-Hydroxygenistein	Isoflavone
pmf0111	253	224.8	6.08	254.06	Anhydroglycinol	Flavonoid
pmf0363	301.1	258	6.28	300.06	Hydroxygenkwanin	Flavonoid
pmf0567	299.1	211	5.72	300.06	Tectorigenin	Flavonoid
pmf0428	407.1	363.1	5.24	408.14	Ginkgolide A	Terpene
pmf0520	283	239	6.58	284.03	Rhein	Quinones
pmf0612	286.1	171.2	7.42	285.14	Piperine	Alkaloids
pmf0299	257.1	240	7.52	256.11	Pterostilbene	Others
pma6625	405.1	243.1	4.63	406.13	E-3,4,5’-Trihydroxy-3’-glucopyranosylstilbene	Others

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
