# Peer review of "UPLC-ESI-MS/MS-Based Widely Targeted Metabolomics Analysis of Wood Metabolites in Teak (Tectona grandis)"

_molecules, 2020, doi:10.3390/molecules25092189_

Round 1

Reviewer 1 Report

Revised manuscript is acceptable.

Author Response

 Thank you for reviewing our manuscript and for the positive response.

Reviewer 2 Report

The manuscript “UPLC-ESI-MS/MS-based widely targeted metabolomics analysis of wood metabolites in teak (Tectona grandis)” by Guang Yang and collaborators provided the primary and secondary metabolome of teak wood.  The comparison among the metabolite compositions within heartwood zone and different sapwood zones was carried out.  It concluded that 16 specifically expressed secondary metabolites might be correlated with the properties of teak heartwood, which could serve as a reference for further study about heartwood formation.

Overall, this study is interesting and worthy of publication in this journal.  There are still some minor revisions as suggested for the authors to improve this manuscript:

Abstract:

1. How to define “excellent properties”?

Introduction:

2. It could be improved with more description describing the targeted metabolomics analysis such as its methodology and related studies.

Materials and Methods:

3. Page 2, line 86: Were the wood samples in the form of cores collected at breast height level (130 cm) in 5 mm diameter representative of the whole wood?

4. Page 3, line 122: The MS analysis was conducted in a positive mode. Should it also be carried in a negative mode to determine more metabolites for metabolomics analysis?

Results

5. It was suggested to show the representative chromatogram in the Result.

6. Page 8, line 263: Please described more detailed how the 16 specially expressed secondary metabolites were screened out.

Author Response

Abstract:

  1. How to define “excellent properties”?

Response: Thank you for pointing out this ambiguity. Accordingly, we have removed the word “excellent” in line 10. We changed “excellent properties” to “durability and color” in line 22.

Introduction:

  1. It could be improved with more description describing the targeted metabolomics analysis such as its methodology and related studies.

Response: According to your suggestion, we have added the following to the the introduction.

Widely targeted metabolomic analysis is a novel approach that combines the advantages of non-targeted metabolomics and targeted metabolomics. Widely targeted metabolomics obtains the potential targets, measured by screening the samples using multiple reaction monitoring (MRM) conditions optimized from the available authentic compounds. Compared to total scan ESI-based non-targeted metabolomics, widely targeted metabolomics based on MRM is a very sensitive and accurate method for the measurement of targeted metabolites.

Materials and Methods

  1. Page 2, line 86: Were the wood samples in the form of cores collected at breast height level (130 cm) in 5 mm diameter representative of the whole wood?

Response: The purpose of this study was to compare the radial horizontal differences in metabolites in teak wood. The height level (130 cm) used is a common location for the investigation of tree growth in forestry, and it is convenient to obtain samples in this manner; thus, we chose to take samples from this location. This height level (130 cm) is not representative of the entire wood. Research on materials at different heights is very important, and we will consider this in the next study.

  1. Page 3, line 122: The MS analysis was conducted in a positive mode. Should it also be carried in a negative mode to determine more metabolites for metabolomics analysis?

Response: Thank you for your suggestion. This was in fact a writing mistake. The MS analysis was conducted in both positive and negative ion mode. In the text, we changed “operating in a positive ion mode” to “operating in both positive and negative ion mode”. Further, in Figure S1, we added the total ion current (TIC) diagram of negative ion mode, TIC stacking diagram of negative ion mode, and multi-peak detection plot of negative ion mode.

Results

  1. It was suggested to show the representative chromatogram in the Result.

Response: Thank you for your suggestion. We have now provided the integral plot of the 16 HWZ-specific metabolites in the results (Figure S2).

  1. Page 8, line 263: Please described more detailed how the 16 specially expressed secondary metabolites were screened out.

Response: We appreciate this suggestion. Based on the expression analysis of secondary metabolites among the comparison groups HWZ vs. ISZ, HWZ vs. MSZ, and HWZ vs. ESZ (Table S2), a total of 16 metabolites could only be detected in heartwood, and these were referred to as HWZ-specific metabolites. We have added this description to the results. Moreover, we have supplemented the integral plot of the 16 HWZ-specific metabolites in the results section (Figure S2).

This manuscript is a resubmission of an earlier submission. The following is a list of the peer review reports and author responses from that submission.

Round 1

Reviewer 1 Report

The manuscript describes a metabolomic study of teak wood Tectona grandis using UHPLC/MS/MS method. While the title of the manuscript is “…targeted metabolomics analysis of wood metabolites…” and the introduction mentioned “Currently 92 metabolites… have been isolated from the root, leaf, and heartwood of teak”, the study reports the identification of 691 metabolites in the wood by MS/MS data. This means this study should be considered as a non-targeted metabolomic analysis. Although the MS/MS function (maybe MSn of QTRAP) may help the structure identification, the application of a low-resolution mass spectrometer like QTRAP MS to do the non-targeted metabolomic analysis is unlikely to be appropriate since this low-resolution MS technique cannot predict a molecular formula of a compound in an acceptable mass accuracy (normally within 5ppm). Therefore, it results in the inaccurate assignments of “125 flavonoids, 100 organic acids and derivatives, 78 amino acids… (Table S1)”. The QTRAP should be used to quantify the levels of the targeted compounds. Except for the structure identification, other chemometric analyses are well-designed and straightforward. I found the manuscript needs to address the structure identification by using high-resolution MS (maybe QTOF, Orbitrap or FT MS) before being accepted for publication in the Molecules journal.

Author Response

Review report 1

Comments and Suggestions for Authors:

The manuscript describes a metabolomic study of teak wood Tectona grandis using uHPLC/MS/MS method. While the title of the manuscript is “…targeted metabolomics analysis of wood metabolites…” and the introduction mentioned “Currently 92 metabolites… have been isolated from the root, leaf, and heartwood of teak”, the study reports the identification of 691 metabolites in the wood by MS/MS data. This means this study should be considered as a non-targeted metabolomic analysis. Although the MS/MS function (maybe MSn of QTRAP) may help the structure identification, the application of a low-resolution mass spectrometer like QTRAP MS to do the non-targeted metabolomic analysis is unlikely to be appropriate since this low-resolution MS technique cannot predict a molecular formula of a compound in an acceptable mass accuracy (normally within 5ppm). Therefore, it results in the inaccurate assignments of “125 flavonoids, 100 organic acids and derivatives, 78 amino acids… (Table S1)”. The QTRAP should be used to quantify the levels of the targeted compounds. Except for the structure identification, other chemometric analyses are well-designed and straightforward. I found the manuscript needs to address the structure identification by using high-resolution MS (maybe QTOF, Orbitrap or FT MS) before being accepted for publication in the Molecules journal

Reply: In this study, we used a widely targeted metabolomic analysis, which is a novel approach that combines the advantages of non-targeted metabolomics and targeted metabolomics.

Compared to the total scan ESI based non-targeted metabolomics (Matsuda et al., 2012), widely targeted metabolomics based on multiple reaction monitoring (MRM) is a very sensitive and accurate method for the measurement of targeted metabolites. The potential targets measured in the current widely targeted metabolomics were obtained by screening the samples using MRM conditions optimized from the available authentic compounds, whereas the endogenous metabolites in the samples were not ‘targeted’ specifically. For endogenous metabolites, MRM transitions (pairs of Q1 and Q3) could be generated from the data gathered in the MS2T library, which was conventionally constructed by total scan ESI-MS/MS (Matsuda et al., 2012). Yao et al. (2008) developed a novel LC-MS/MS method that uses multiple ion monitoring (MIM) as a survey scan to trigger the acquisition of enhanced product ions (EPI) to identify drug metabolites. The MIM-EPI has been shown to be suitable for obtaining the fragmentation patterns of a large number of metabolites with similar sensitivity to MRM-EPI, and is higher than that of the total scan EMS-EPI. Moreover, since Q3=Q1 was set in MIM-EPI, MS2 data could be obtained for unknown metabolites and/or metabolite with unpredictable fragmentation patterns, which is impossible for MRM-EPI (Chen et al., 2013).

We adopted a strategy known as stepwise MIM–EPI to build the commercially available standard Metabolites Database. The main processes of building the Metabolites Database are shown in Figure 1 (Chen et al., 2013).

Figure 1. The Main Procedures for Stepwise MIM–EPI (Multiple Ion Monitoring–Enhanced Product Ions)-Based MS2T Library Construction and MRM (Multiple Reaction Monitoring)-Based Widely Targeted Metabolic Profiling.

Then we matched the metabolite mass spectrometry information including Q1, Q3, retention time (RT), collision energy (DP), and de-clustering potential (CE) obtained from the experimental samples with the Metabolites Database to identify the metabolites.

References:

[1] Chen et al. (2013). A Novel Integrated Method for Large-Scale Detection, Identification, and Quantification of Widely Targeted Metabolites: Application in the Study of Rice Metabolomics. Molecular Plant. 6 ,1769–1780.

[2] Matsuda et al. (2012). Dissection of genotype–phenotype associations in rice grains using metabolome quantitative trait loci analysis. Plant J. 70, 624–636.

[3] Yao et al. (2008). Rapid screening and characterization of drug metabolites using a multiple ion monitoring-dependent MS/MS acquisition method on a hybrid triple quadrupole-linear ion trap mass spectrometer. J. Mass Spectrom. 43, 1364–1375.

Reviewer 2 Report

Information of matching the metabolites to existing 'Metware database' by tandem LC-MS was not sufficient.

Metabolites matching using LC-MS/MS, not using TOF MS, should be concretely presented (it may be provided as a supplementary material).

But, 'Supplementary Materials' were not attached in review system. Please check.

Author Response

Review report 2

Comments and Suggestions for Authors:

Information of matching the metabolites to existing 'Metware database' by tandem LC-MS was not sufficient.

Metabolites matching using LC-MS/MS, not using TOF MS, should be concretely presented (it may be provided as a supplementary material).

But, 'Supplementary Materials' were not attached in review system. Please check.

Reply: We matched the metabolite mass spectrometry information including Q1, Q3, retention time (RT), collision energy (DP), and de-clustering potential (CE) and showed the information of Q1, Q3, and RT in the supplementary material (Table S1). We were not able to provide the DP and CE information owing to the confidentiality rules of the company.

Round 2

Reviewer 1 Report

The authors gave several evidence with references to support the MS/MS method they approached. With their current database, their non-targeted metabolomic study using QQQ-MS is convincible. I would recommend the revised manuscript should be accepted to be published.

Reviewer 2 Report

Regarding to Table S1, all mass spectra of each detected metabolite should be provided by its matched result with the database by separate supplementary figures.  Moreover, the information about MetWare database should be detailed.